# F-Box/WD Repeat Domain-Containing 7 Induces Chemotherapy Resistance in Colorectal Cancer Stem Cells

**DOI:** 10.3390/cancers11050635

**Published:** 2019-05-07

**Authors:** Shusaku Honma, Shigeo Hisamori, Aya Nishiuchi, Yoshiro Itatani, Kazutaka Obama, Yohei Shimono, Yoshiharu Sakai

**Affiliations:** 1Department of Surgery, Graduate School of Medicine, Kyoto University, Kyoto 606-8507, Japan; shomma74@kuhp.kyoto-u.ac.jp (S.H.); annya@kuhp.kyoto-u.ac.jp (A.N.); itatani@kuhp.kyoto-u.ac.jp (Y.I.); kobama@kuhp.kyoto-u.ac.jp (K.O.); ysakai@kuhp.kyoto-u.ac.jp (Y.S.); 2Department of Biochemistry, School of Medicine, Fujita Health University, Aichi 470-1192, Japan; yshimono@fujita-hu.ac.jp

**Keywords:** FBXW7, colorectal cancer, cancer stem cell, chemoresistance, cell cycle

## Abstract

Although the cancer stem cell (CSC) concept has provided a reasonable explanation for cancer recurrence following chemotherapy, the relationship between CSCs and chemotherapy resistance has not been thoroughly investigated, especially in solid tumors. We aimed to identify the mechanism underlying colorectal cancer (CRC) chemoresistance focusing on the cell cycle mediator F-Box/WD repeat domain-containing 7 (FBXW7). From 55 consecutive CRC cases who underwent neoadjuvant chemotherapy (NAC) or neoadjuvant chemoradiotherapy (NACRT) at Kyoto University Hospital, pre-treatment endoscopic biopsy specimens were collected and divided into two groups upon immunohistochemical (IHC) analysis: 21 cases of FBXW7 high expression (FBXW7-high group) and 34 cases of low expression (FBXW7-low group). High FBXW7 expression in pre-treatment biopsy specimen was significantly associated with poor pathological therapeutic effect (*p* = 0.019). The proportion of FBXW7-positive cells in surgically resected CRC specimens from patients who underwent NAC or NACRT was significantly higher than that in the pre-treatment biopsy specimens (*p* < 0.001). The expression of FBXW7 was inversely correlated with that of Ki67 in both pre-treatment biopsy specimens and surgically resected specimens. *FBXW7* expression in the EpCAM^high^/CD44^high^ subpopulation isolated by flow cytometry from CRC samples was significantly higher than that in the EpCAM^high^/CD44^low^ subpopulation. Cell-cycle analysis in CRC cell lines revealed that, upon *FBXW7* silencing, the proportion of G0/G1 cells was significantly lower than that in control cells. Moreover, knockdown of *FBXW7* in CRC cell lines increased the sensitivity to anti-cancer drugs in vitro and in vivo. A subset of CRC stem cells possesses chemoresistance through FBXW7 expression. Cell cycle arrest induced by FBXW7 expression should be considered as a potential therapeutic target to overcome chemoresistance in CRC stem cell subsets.

## 1. Introduction

Colorectal cancer (CRC) is one of the most common causes of cancer deaths worldwide due to its recurrence and chemoresistanse [1]. Multidisciplinary treatment with chemoradiotherapy (CRT) followed by surgery, which allows an early attack on systemic micro-metastases and downstaging of the primary tumor, is the standard strategy for locally advanced rectal cancer to increase the possibility of curative resection [2,3]. On the other hand, adjuvant chemotherapy following surgery is commonly performed for advanced CRC [4]. Although most of patients with CRC achieve clinical remission through combined modality therapies, a substantial proportion of them experiences recurrence and chemoresistance, resulting in poor prognosis.

Recent studies have suggested that tumors show cellular hierarchy with a subpopulation of cancer cells that can renew and generate differentiated tumor cells [5,6,7,8]. This highly tumorigenic cells have been defined as cancer stem cells (CSCs), conventionally sorted based on the expression of several cell surface markers such as CD44, CD133 and CD166 [9,10,11,12]. The CSC theory provides a plausible explanation for chemoresistance and cancer relapse. Accumulating evidence has revealed that CSCs are generally resistant to various therapeutic interventions through their quiescence, capacity for DNA repair and ATP-binding cassette drug transporter expression [13,14]. Especially, the ability to maintain quiescence, which is known as dormancy, is responsible for chemoresistance, since conventional anti-cancer therapies preferentially target dividing cells [15]. Failure to eliminate quiescent CSCs may result in regrowth or clinical recurrence of the tumor. Therefore, the development of therapeutic approaches that target quiescent CSCs is necessary to eradicate cancers. 

F-box and WD repeat domain containing 7 (FBXW7), a vital component of the SKP1, CUL1 and F-box protein (SCF)-type ubiquitin ligase complexes that ubiquitinate specific substrates, is a crucial cell cycle regulator, which acts through the degradation of cell-cycle accelerators such as cyclin E, c-Myc, c-Jun, Notch and MCL1 [16,17,18,19]. Since these cell-cycle accelerators are known as oncoproteins that are frequently upregulated in a wide range of human cancers including CRC, FBXW7 has been initially considered a tumor suppressor [20,21]. On the other hand, a previous study has shown that chronic myeloid leukemia (CML)-initiating cells (LICs) express high levels of FBXW7 and are maintained in a quiescent state; interestingly, abrogation of the cell-cycle quiescence through *FBXW7* depletion makes LICs more sensitive to imatinib, commonly used to treat CML [22]. Moreover, a recent report using specific CRC stem cell lines has suggested that CRC stem cells could acquire chemoresistance by upregulating FBXW7 and becoming quiescent via c-Myc degradation [23]. Therefore, when focusing on the elimination of all cancer cells, the role of FBXW7 in CRCs is still controversial. 

In the present study, we investigated the relationship between FBXW7 expression and chemotherapeutic efficacy in primary CRC. 

## 2. Results

### 2.1. High FBXW7 Expression in Pre-Treatment Biopsy Specimens Is Related to Poor Pathological Theraperutic Effect

We first investigated the relationship between FBXW7 expression in the pre-treatment biopsy specimens and the pathological therapeutic effect of NAC/NACRT followed by surgical resection. We characterized FBXW7 expression in the pre-treatment biopsy specimens as low (IHC score, 1, 2) or high (IHC score, 3, 4, 6; Figure 1A). The association between the patient clinicopathological features and FBXW7 expression is summarized in Table 1. Fifty-five CRC cases were divided into 21 cases of high FBXW7 expression and 34 cases of low FBXW7 expression. We found that high FBXW7 expression in pre-treatment biopsy specimen was significantly associated with poor pathological therapeutic effect (Figure 1B). However, no significant association was observed between FBXW7 expression and age, gender, tumor location, chemotherapy regimen, use of molecular target drug, radiation therapy, histology, clinical N status, clinical M status, or clinical stage.

### 2.2. FBXW7 Is Highly Expressed in the Post-Treatement Resected Specimens

Next, we analyzed the expression of FBXW7 in the post-treatment resected specimens from the patients who underwent NAC/NACRT followed by surgery and compared it with FBXW7 expression in the pre-treatment biopsy specimens. Samples from six of the 55 patients described above could not be evaluated (e.g., because of pathologically grade 3 complete remission after NAC/NACRT); therefore, we excluded these six patients from further analyses. Figure 2A shows the data from two patients: patient #1, with high expression of FBXW7, showed low pathological therapeutic effect and the proportion of FBXW7 -positive cells in the post-treatment resected specimens was higher than in the pre-treatment biopsy specimens. On the other hand, patient #2, with low expression of FBXW7, showed high pathological therapeutic effect: most of the tumor cells died after treatment and almost all of residual tumor cells were FBXW7-positive (Figure 2A). The timeline between the end of latest NAC/NACRT and operation of 55 CRC patients was 7.6 ± 3.8 week (mean ± standard deviation (SD)). 

We compared FBXW7 staining score in pre-treatment biopsy specimens and post-treatment resected specimens and found that, in most of the cases, it was relatively higher in the latter (Figure 2B). There was no case in which the score was lower in the post-treatment resected specimen. Table 2 shows FBXW7 staining scores in pre-treatment biopsy specimens and post-treatment resected specimens for each patient. 

We also performed IHC staining of Ki-67 in the specimens. In the pre-treatment biopsy specimens, the proportion of Ki-67-positive cells was low when FBXW7 was high, whereas the proportion of Ki-67-positive cells was high when FBXW7 was low (Figure 2C). The residual tumor cells after treatment were FBXW7-positive and Ki-67-negative in both FBXW7-high and -low cases. We found that the expression of FBXW7 was inversely correlated with that of Ki67 in both pre-treatment biopsy specimens and post-treatment resected specimens. In addition, we showed the representative pictures of these 4 cases with the staining of H&E, FBXW7, and Ki-67 (Appendix A). We measured the percentage of Ki-67-positive cells from five independent visual fields with high-magnifications for each case, and could reveal that the number of Ki-67-expressed cells tended to be reduced in post-treatment resected specimens compared to that in pre-treatment biopsy specimens.

### 2.3. FBXW7 Expression in the EpCAM^high^/CD44^high^ Population Was Significantly Higher Than That in The EpCAM^high^/CD44^low^ Population

According to the clinical data described above, we hypothesized that FBXW7 was correlated with the stemness of CRC cells. CRC stem cells have been shown to be EpCAM^high^/CD44^high^ [9]. We divided tumor cells from human CRC xenograft tumors (patient-derieved xenografts (PDXs)) into the EpCAM^high^/CD44^high^ and EpCAM^high^/CD44^low^ populations (Figure 3A) and compared their *FBXW7* expression by quantitative PCR (qPCR). In all four PDXs, the EpCAM^high^/CD44^high^ population showed significantly high *FBXW7* expression compared with the EpCAM^high^/CD44^low^ population (Figure 3B). 

We also analyzed the mRNA expression levels of BMI1 and LGR5, known as intestinal stem cell markers. Two PDXs showed significantly higher BMI1 expression in the EpCAM^high^/CD44^high^ population than in the EpCAM^high^/CD44^low^ population, while there were no differences in the other two PDXs (Appendix A). In three PDXs, the levels of LGR5 in the EpCAM^high^/CD44^high^ population was significantly higher than that in the EpCAM^high^/CD44^low^ population, whereas no significant difference was seen in the remaining one.

### 2.4. FBXW7 Knockdown Accelerates Cell cycle and Cell Proliferation, Resulting in Increased Sensitivity to Anti-Cancer Drugs in CRC Cells

Our data in clinical samples suggested that FBXW7 played an important role in the maintenance of CRC stemness and resistance to chemotherapy/CRT. To evaluate the effect of FBXW7 silencing on CRC cell lines, we established stably *FBXW7* knocked-down CRC cell lines through infection of lentiviral particles expressing shRNAs. To this end, two distinct shRNAs targeting *FBXW7* were used. Knockdown of FBXW7 was confirmed by qPCR (mRNA level), flow cytometry and Western blotting analyses (protein level) (Appendix A). To address the effect of FBXW7 on cell-cycle regulation, we performed cell cycle analysis with cell synchronization by double thymidine block. In the three CRC cell lines analyzed, the percentage of G0/G1 phase in sh*FBXW7* cells was significantly lower than that in scrambled control cells (Figure 4A). Notably, the percentage of cells in S and G2/M in sh*FBXW7* cells was significantly higher than that in scrambled HT29 control cells, while the fraction of cells in S phase and G2/M phase in the sh*FBXW7* population was significantly higher than that in scrambled control SW480 and DLD-1 cells, respectively (Figure 4A). These results suggested that FBXW7 plays a critical role in the maintenance of quiescence (G0/G1 phase) in CRC cells. 

Furthermore, we evaluated the proliferation of the three CRC cell lines upon FBXW7 stable knockdown. We found that the proliferation rates in all three CRC cell lines with FBXW7 stably knocked-down was significantly higher than that in scrambled control cells (Figure 4B). Therefore, FBXW7 knockdown accelerated cell cycle and proliferation. 

To investigate whether FBXW7 affected the sensitivity of the cells to anti-cancer drugs is preferentially effective on proliferative cells, we performed cytotoxicity assays on the three CRC cell lines using 5-FU, L-OHP, or CPT-11, chemotherapeutic drugs that are frequently used for clinical treatment of CRC. We found that FBXW7 knockdown significantly increased the sensitivity to these anti-cancer drugs in the three CRC cell lines examined (Figure 4C). 

### 2.5. FBXW7 Knockdown Inhibits Tumor Growth Rates upon Chemotherapy Treatment in Vivo

We further examined the effect of *FBXW7* knockdown on the sensitivity to anti-cancer drugs using xenograft models. We used DLD-1 cells treated with CPT-11 for further experiments. Representative images of the tumors from mice inoculated with scrambled or *FBXW7*-silenced cells are shown in Figure 5A. In the mock-treated and sh*FBXW7* #1-expressing group, tumors grew significantly faster than in the mock-treated and scrambled control expressing group. Although there was no significant difference, tumors from the mock-treated sh*FBXW7* #2-expressing group tended to grow faster than mock-treated and scrambled control expressing group (Figure 5B,C). Notably, in the CPT-11-treated mice, there were no significant differences in the tumor growth from the scrambled control and *FBXW7*-silenced group (Figure 5C), whereas the tumor growth inhibition rates were significantly higher in tumors from mice inoculated with sh*FBXW7* cells than those from mice inoculated with scrambled control cells (Figure 5D).

## 3. Discussion

Chemoresistance is the major limitation in the survival and management of patients with CRC. Increasing evidence has suggested that a minor subset of cancer cells obtains chemoresistance through dormancy, leading to the clinical failure of treatment [13,24,25]. EpCAM^high^/CD44^high^ cells have been proposed as colorectal CSCs [9,26]. The results of our study indicated that a subset of EpCAM^high^/CD44^high^ population was dormant and chemoresistant while maintaining FBXW7 expression. We hypothesized that colorectal CSC population sorted with cell surface markers such as CD44 and revealed to be tumorigenic is still heterogeneous and, in a part, contains a subset of dormant cells responsible for the clinical recurrence and chemoresistance. According to our data, FBXW7 expression could be a significant factor negatively interfering with complete remission after CRT. 

As to the proportion of FBXW7-positive cells in patients with CRC, we found that in surgically resected specimens following NAC/NACRT this proportion was relatively higher than that in the pre-treatment biopsy specimens. Additionally, IHC analysis showed that high FBXW7 staining score in the pre-treatment biopsy specimens was significantly associated with the poor pathological therapeutic effect. These results not only suggest that FBXW7-expressing CRC cells are chemoresistant but also indicate that the proportion of FBXW7-positive cells in pre-treatment biopsy specimens might be a predictor of the response to NAC/NACRT. 

Recent studies have shown that loss of FBXW7 function causes therapeutic resistance through the accumulation of several oncoproteins associated with tumor progression and therapeutic resistance, including cyclin E, c-Myc, c-Jun, Notch and MCL1 [27,28,29,30,31]. Accordingly, in in vitro experiments we found that *FBXW7*-silenced CRC cells were associated with cell cycle progression and cell proliferation and, in in vivo experiments, we demonstrated that the tumor generated upon injection of *FBXW7*-silenced cells were more aggressive than those generated from scrambled control cells. On the other hand, the growth inhibition rate upon treatment with CPT-11 was significantly higher in tumors from *FBXW7*-silenced cells than in those from scrambled control. These findings suggest that suppression of FBXW7 function would provide a therapeutic window in clinical settings. This would also be in line with the previously published work in the field, as accelerating tumor cells should make them more vulnerable [22]. Therefore, focusing on the eradication of CRC through complete pathological remission, FBXW7, which induces chemo-resistance in CRC, could be a significant therapeutic target. 

The current study has some limitations. First, our exploratory results were obtained from samples retrospectively evaluated and collected in a single institute; the sample size was relatively small, although we could show statistical significance. Second, we evaluated the therapeutic effect of preoperative therapy only analyzing the pathology of the post-treatment resected specimens. A comprehensive assessment of the preoperative therapy including Response Evaluation Criteria in Solid Tumors (RECIST) to evaluate the relationship between FBXW7 expression and the preoperative therapy might have been more informative in this regard. Third, a significant number of the patients included in this study were diagnosed with rectal cancer whereas we used CRC cell lines for the experiments. Nevertheless, we successfully demonstrated that high expression of FBXW7 was associated with chemoresistance in multiple CRC cell lines, both in vitro and in vivo, supporting our clinical findings. In addition, we could not investigate the behavior of *FBXW7*-mutated CRC cells in this project. However, the mutation frequency of *FBXW7* in CRC has been reported as about 11% in non-hypermutated samples [32]. We have spared the small amount of *FBXW7*-mutated cases, instead, we could suggest that the modification of FBXW7 expression might be a new strategy for CRC treatment. It is possible to consider that FBXW7 would be regulated in a post-transcriptional manner. 

Although we could successfully show that suppression of FBXW7 by lentiviral stable knockdown dramatically induced chemosensitivity in vitro and in vivo, so far there is no available drug that specifically suppresses FBXW7 function. Moreover, simple knockdown of *FBXW7* could promote tumor progression (Figure 4B and Figure 5A–C). Therefore, even if an inhibitor of FBXW7 might be developed, it could become a “double-edged sword” in a real clinical setting when used as a single agent. Finally, the colorectal CSC population might be more heterogeneous than we expect; therefore, combination therapy to target the dormant cancer cells as well as the proliferating progeny should be considered. 

In summary, we believe that a subset of colorectal CSCs maintains chemoresistance induced by the expression of FBXW7, which should be considered as a potential therapeutic target to overcome chemoresistance in a colorectal CSC subset. 

## 4. Materials and Methods 

### 4.1. Patient Population

For the clinicopathological analysis, biopsy specimens and resected specimens were obtained from 55 patients with CRC who received chemotherapy or CRT before surgical resection at Kyoto University Hospital (Kyoto, Japan) between January 2010 and December 2017. The medical records of all patients were retrospectively reviewed. The diagnosis of CRC was confirmed by pathological examination. The pathological therapeutic effects of preoperative treatment are defined in the Japanese Classification of Colorectal Carcinoma, 8th Edition [33] as follows: grade 0 = ineffective; grade 1 = mildly effective; grade 2 = significantly effective; grade 3 = markedly effective without viable cancer cells. In this study, grade 0, 1 and 2, 3 were defined as a poor and great pathological therapeutic effect, respectively. This study protocol was approved by the institutional review board of Kyoto University (confirmation #R1251), and patients provided their consents for the sample use and data analysis.

### 4.2. Cell Lines and Reagents

The CRC cell lines HT29-luc and DLD-1-luc cells, stably expressing firefly luciferase, were obtained from JCRB Cell Bank (Osaka, Japan) and SW480 was obtained from the American Type Culture Collection (Manassas, VA, USA). Stable transductants of SW480 cells with firefly luciferase were established using luciferase sequence in the pGL4.10 vector (Promega, Madison, WI, USA) as previously described [34]. The identity of all cell lines was confirmed by short tandem repeat analysis. Cells were cultured in high glucose DMEM (Wako, Tokyo, Japan) with 10% fetal bovine serum (Life Technologies Japan, Tokyo, Japan) and penicillin (100 U/mL) and streptomycin (100 µg/mL; Life Technologies Japan), and were incubated at 37 °C in a humidified chamber containing 5% CO_2_. shRNA-expressing lentiviruses were produced by transfecting HEK293T cells with human *FBXW7* shRNA constructs in lentiviral GFP vectors (TL304541, OriGene Technologies, Rockville, MD, USA) together with the packaging vector psPAX2 and the envelope vector pMD2.G using Lipofectamine 2000 (Thermo Fisher Scientific, Waltham, MA, USA). CRC cells were infected with the lentiviruses in the presence of polybrene; stable knockdown cells were selected by flow cytometry, collecting GFP-positive cells.

### 4.3. IHC Studies

IHC studies for FBXW7 and Ki-67 were performed on formalin-fixed, paraffin-embedded sections obtained from CRC tissues. All of these samples used in this study were diagnosed by two independent pathologists as adenocarcinoma, CRC origin. Tissue sections were deparaffinized and pre-treated for antigen retrieval by autoclave heating in 10 mM sodium citrate buffer (pH 6.0) for 20 min. Endogenous peroxidase activity was blocked using 3% hydrogen peroxide in methanol for 15 min. The sections were incubated in primary mouse anti-human FBXW7 monoclonal antibody (3 µg/mL, clone 3D1, Abnova Corporation, Taipei, Taiwan) at 4 °C overnight or mouse anti-human Ki-67 monoclonal antibody (1:100 dilution, clone MIB-1, Dako, Glostrup, Denmark) at 20–25 °C for 20 min. After washing with PBS, the sections were incubated for 1 hour in goat anti-mouse secondary antibody conjugated to biotin (1:200 dilution, Thermo Fisher Scientific) at 20–25 °C. Immunoreactive complexes were visualizes with the avidin-biotin immunoperoxidase method and using a diaminobenzidine solution. All sections were counterstained with Gill’s hematoxylin. We confirmed the sensitivity and specificity of the FBXW7 antibody using xenograft tumors derived from WT (stably expressing control shRNA) or stably *FBXW7* silenced DLD-1 cells (Appendix A). The expression status of FBXW7 was scored according to the percentage of positive cells and staining intensity. Scoring for percentages of positive staining were: 1 for 0–50%, 2 for 51–75%, and 3 for more than 76% stained cells. Scoring for staining intensity was: 1 for absent to moderate staining, and 2 for strong staining. These values were multiplied to characterize FBXW7 expression as low (1, 2) or high (3, 4, 6) ranging (Appendix A). 

### 4.4. Xenograft Analysis

Human CRC xenograft tumors (patient-derived xenografts; PDXs) were mechanically and enzymatically dissociated into single-cell suspensions using a Human Tumor Dissociation Kit (Miltenyi Biotec, Bergisch Gladbach, Germany), according to the manufacturer’s instructions. The single-cell suspensions from the PDXs were immunohistologically stained with a mouse anti-human EpCAM antibody (clone EBA-1, Becton, Dickinson and Company (BD), Franklin Lakes, NJ, USA), a mouse anti-human CD44 antibody (clone G44-26, BD). After washing, the cells were resuspended with PBS containing 1% bovine serum albumin, and dead cells were labeled with 1.0 µg/mL 4’,6-Diamidino-2-Phenylindole, Dihydrochloride (DAPI; Life Technologies Japan). These samples were analyzed using a FACS Aria II cell sorter (BD).

### 4.5. RNA Isolation and qPCR

Total RNA from cultured CRC cells and sorted xenograft tumor cells was extracted using the High Pure RNA Isolation Kit (Roche Diagnostics, Indianapolis, IN, USA) and miRNeasy Micro Kit (Qiagen, Tokyo, Japan), respectively, according to the manufacture’s protocols. Complementary DNAs was prepared using ReverTra ACE (TOYOBO Life Science, Osaka, Japan) and qPCR assays were performed using SYBR Green reagents (TOYOBO Life Science) in an ABI Step One Plus thermal cycler (Thermo Fisher Scientific). The primer sequences used for qPCRs are listed in Appendix A. We measured the expression of all three *FBXW7* isoforms. The mRNA level of each gene was normalized to that of *ACTB*.

### 4.6. Cell Synchronization and Cell Cycle Analysis

CRC cells were synchronized at late G1/early S using the double thymidine block method as previously described [35]. Cells were collected 0 and 24 h after release. Cells were dissociated with trypsin/EDTA, washed with PBS, and fixed with ice-cold 70% ethanol for 30 min. Cells were then resuspended in PI/RNase Staining Buffer (BD) and incubated for 15 min at 20–25 °C. These samples were analyzed using a FACS Aria II cell sorter. Cell cycle profiles were analyzed using the FlowJo (version 10) data analysis software (BD) to determine the percentage of cells in G0/G1, S and G2/M.

### 4.7. Luminescence-Based Proliferation and Cytotoxicity Assays

CRC cell lines (5–10 × 10^3^ cells/well) were seeded in wells of Nunc F96 Microwell white plates (Thermo Fisher Scientific) in a final volume of 100 µL culture medium containing 200 µg/mL D-luciferin (Promega) per well. After over-night incubation, luminescence was scored using a GloMax-Multi detection system (Promega). This initial measurement was defined as the day 0-value; luminescence was then determined daily. The cell proliferation rate for each cell line was estimated as the ratio of photon count at the chosen timepoint to that on day 0 [36]. For the cytotoxicity assay, CRC cell lines (5–10 × 10^3^ cells/well) were seeded in 96-well white cell culture plates. After over-night incubation, irinotecan (CPT-11; Wako), oxaliplatin (L-OHP; Wako) or 5-fluorouracil (5-FU; Wako) were added at the indicated concentration and luminescence was determined at 24, 48, 72 h after administration of anti-cancer drugs. Cell viability was calculated as the treated/control cell ratio of photon counts. 

### 4.8. FBXW7 Protein Analysis

CRC cells were dissociated with trypsin/EDTA, washed with PBS, and fixed with 4% paraformaldehyde. The cells were permeabilized using ice-cold methanol for 30 min. After washing, nonspecific antibody binding sites were blocked with 5% normal goat serum/PBS for 30 min. Cells were then incubated with rabbit anti-human FBXW7 monoclonal antibody (1:200 dilution, clone SP237, Abcam, Cambridge, UK) or rabbit IgG isotype control (Abcam) for 30 min at 4 °C, followed by APC conjugated goat anti-rabbit IgG secondary antibody (Abcam) for 30 min at room temperature. These samples were analyzed using a FACS Aria II cell sorter.

### 4.9. Western Blotting Analysis

A total of 20 μg of whole cell lysate were subjected to sodium dodecyl sulfate polyacrylamide gel electrophoresis and transferred to polyvinylidene difluoride membrane (Merck Millipore, Darmstadt, Germany), as previously described [37]. Membranes were immunoblotted with anti-human FBXW7 monoclonal antibody (1:1000 dilution, clone SP237, Abcam) followed by horseradish peroxidase (HRP)-conjugated secondary antibody (1:4000 dilution, Dako). HRP-conjugated ACTB antibody (1:8000 dilution, clone AC-15, Sigma-Aldrich, St. Louis, MO, USA) was used as a loading control.

### 4.10. In vivo Xenograft Studies

A total of 1 × 10^6^ control (stably expressing control shRNA) or stably *FBXW7* silenced DLD-1 cells were suspended in 50 µL PBS and an equal volume of Matrigel (BD) and subcutaneously injected into both dorsal flanks of 6–7-week-old female BALB/c nude mice (CLEA Japan, Osaka, Japan)—three mice for each situation, that was, we used 18 mice in this experiment. Nine mice were treated with intraperitoneal injection of CPT-11 (10 mg/kg, every four days) and the other nine mice were mock-treated. Treatment began on the day after inoculation. Tumor sizes were measured with calipers twice a week and the tumor volumes were calculated with the formula *V* = 0.5 × *L* × *W*^2^, where *V* = volume, *L* = length, *W* = width (with length greater than width). The tumor growth inhibition rate was calculated as the percentage of the reduction of CPT-11-treated tumor volume in comparison with the tumor volumes in the mock-treated. Tumors from mice in which troubles during inoculation, such as cell suspension leakage from the needle hole soon after injection or intradermal injection, had arisen were excluded from the evaluation. Tumors from a mouse that died after CPT-11 misdirected administration were also excluded. The animal experiment was approved by the Animal Care and Use Committee of Kyoto University (Medkyo18150).

### 4.11. Statistical Analysis

All values are expressed as means ± standard deviation. All in vitro experiments were performed at least three times. The statistical significance of differences was determined by Mann-Whitney *U* test, Wilcoxon *t*-test, Fisher’s exact test or Student’s *t*-test, as appropriate. All analyses were two-sided, and a *p* value <0.05 was considered to be statistically significant.

## 5. Conclusions

A subset of CRC stem cells possesses chemoresistance through FBXW7 expression. Cell cycle arrest induced by FBXW7 expression should be considered as a potential therapeutic target to overcome chemoresistance in CRC stem cell subsets.

## Figures and Tables

**Figure 1 cancers-11-00635-f001:**
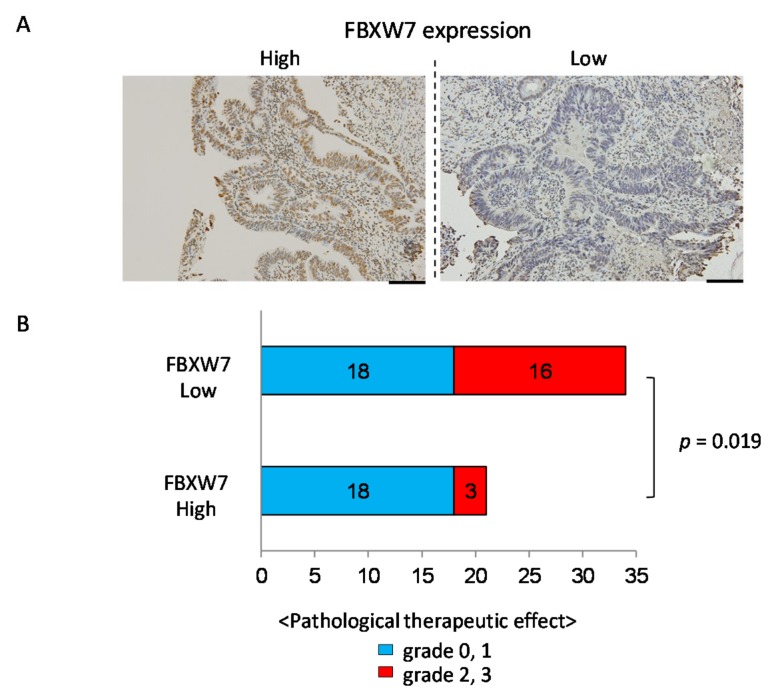
High FBXW7 expression in pre-treatment biopsy specimens is related to poor pathological therapeutic effect. (**A**) IHC staining for FBXW7 in representative pre-treatment biopsy specimens. Left panel shows high FBXW7 expression and right panel shows low FBXW7 expression. Scale bars, 100 µm. (**B**) Correlation between FBXW7 expression in pre-treatment biopsy specimens and the pathological therapeutic effect of NAC/NACRT in surgically resected specimens. X-axis, number of cases.

**Figure 2 cancers-11-00635-f002:**
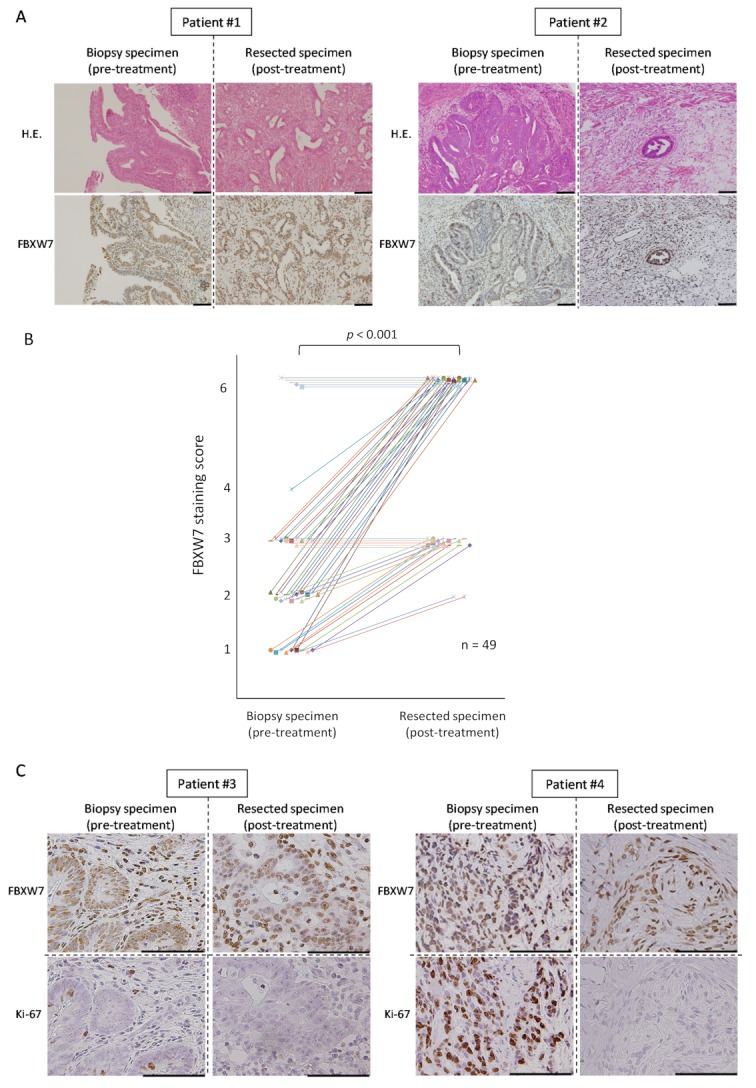
FBXW7 expression in the post-treatment resected specimens is higher than that in the pre-treatment biopsy specimens. (**A**) Hematoxylin and eosin (H.E.) and IHC staining for FBXW7 in serial sections of representative CRC specimens. Patient #1 belongs to the FBXW7-high group and shows low pathological therapeutic effect. Patient #2 belongs to the FBXW7-low group and shows high pathological therapeutic effect. Scale bars, 100 µm. (**B**) Comparison of FBXW7 staining score between pre-treatment biopsy specimens and post-treatment resected specimens. We evaluated 49 cases in which FBXW7 IHC staining could be performed on both biopsy and surgically resected specimens. (**C**) IHC staining for FBXW7 and Ki-67 in serial sections of representative CRC specimens. Patient #3 belongs to the FBXW7-high group and shows low pathological therapeutic effect. Patient #4 belongs to the FBXW7-low group and shows high pathological therapeutic effect. Scale bars, 100 µm.

**Figure 3 cancers-11-00635-f003:**
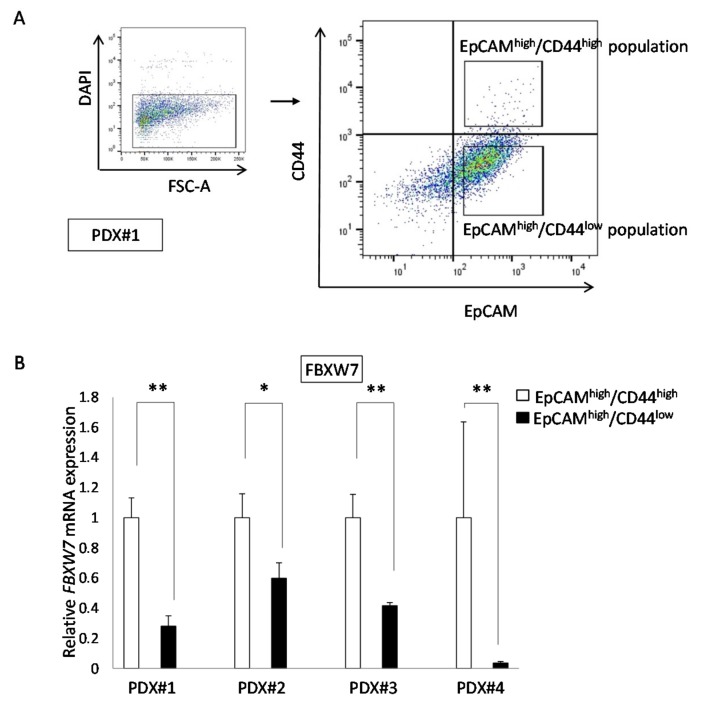
Analysis of human CRC PDXs. (**A**) Representative flow cytometric plot. The EpCAM^high^/CD44^high^ and EpCAM^high^/CD44^low^ populations were collected by flow cytometry. (**B**) *FBXW7* expression in the four types of PDXs. *FBXW7* expression in the EpCAM^high^/CD44^high^ population was significantly higher than that in the EpCAM^high^/CD44^low^ population in all four PDXs. Results are presented as the means ± standard deviation of at least three independent experiments. * *p* < 0.05, ** *p* < 0.01.

**Figure 4 cancers-11-00635-f004:**
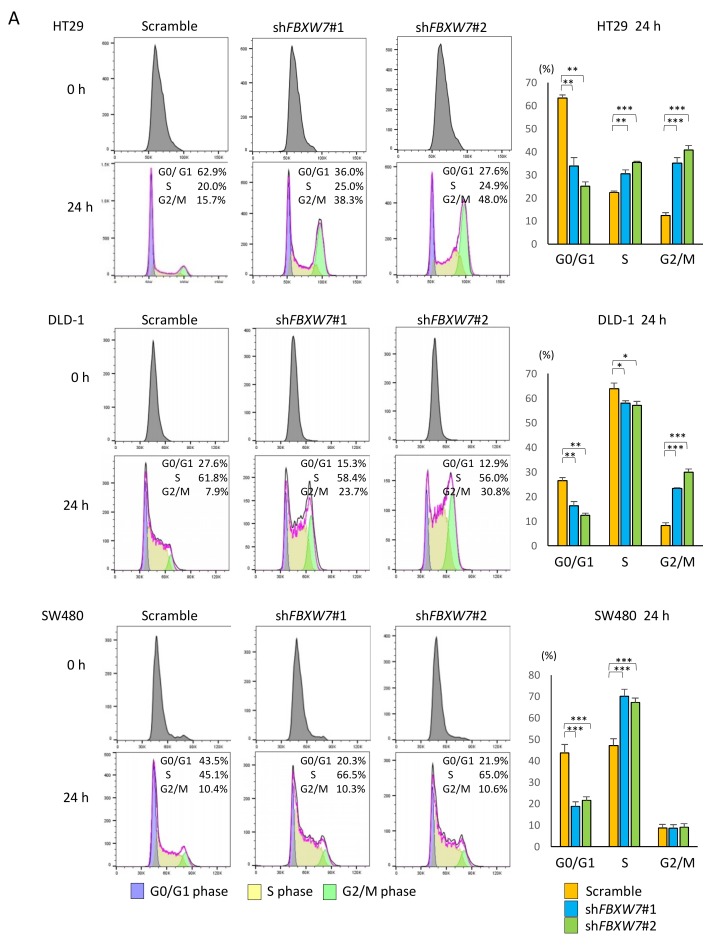
*FBXW7* knockdown accelerates cell cycle and cell proliferation and increases sensitivity to anti-cancer drugs in CRC cell lines. (**A**) Cell cycle distributions in control and *FBXW7*-silenced cells following release from cell synchronization by double thymidine block. Upper and lower panels for each CRC cell line show cell cycle distribution at 0 and 24 h after release from double thymidine synchronization. Bar graphs indicate the percentage of cells in G0/G1, S or G2/M phase 24 h after release from cell synchronization. (**B**) Luminescence-based cell proliferation assays in luciferase-expressing CRC cell lines in which *FBXW7* was stably knocked down. (**C**) Luminescence-based cytotoxicity assays in luciferase-expressing CRC cell lines in which *FBXW7* was stably knocked down. Cells were incubated for 24, 48 or 72 h with the indicated concentrations of anti-cancer drugs. Cell viability was calculated as the treated/control cell ratio of photon counts. Results are presented as the means ± standard deviation of at least three independent experiments. * *p* < 0.05, ** *p* < 0.01, *** *p* < 0.001.

**Figure 5 cancers-11-00635-f005:**
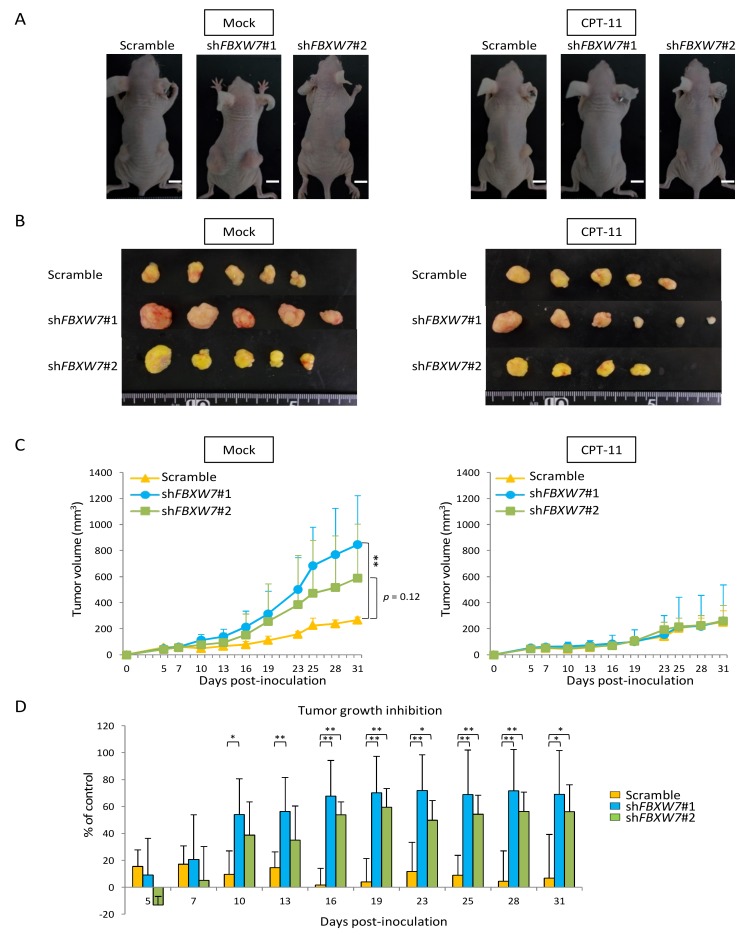
In a xenograft model, *FBXW7* knockdown promotes tumor growth and induces tumor inhibition upon treatment with CPT-11. (**A**) Representative images of tumors in nude mice inoculated with control or FBXW7-silenced DLD-1 cells, treated or not with CPT-11, 31 days after inoculation. White scale bars, 1 cm. (**B**) Macroscopic images of the tumors in nude mice inoculated with control or *FBXW7*-silenced DLD-1 cells, treated or not with CPT-11, 31 days after inoculation. (**C**) Tumor growth curves of xenografts derived mice inoculated with control or *FBXW7*-silenced DLD-1 cells, treated or not with CPT-11. (D) Tumor growth inhibition rates. Results are presented as the means ± standard deviation of at least three independent experiments. * *p* < 0.05, ** *p* < 0.01, *** *p* < 0.001.

**Table 1 cancers-11-00635-t001:** Correlation between FBXW7 expression and clinicopathological features in 55 patients with CRC treated with NAC/NACRT before surgical resection.

Characteristics	FBXW7 Expression			
	Total	High	Low	*p* Value
	n = 55	n = 21	n = 34	
Age				0.464
mean ± SD		60.6 ± 13.4	62.6 ± 7.0	
range		26–79	50–73	
Gender				0.956
Male	43	16	27	
Female	12	5	7	
Location				0.684
Right hemicolon	2	2	0	
Left hemicolon	1	1	0	
Sigmoid colon	7	3	4	
Rectum	45	15	30	
Chemotherapy regimen				0.98
L-OHP base	40	16	24	
CPT-11 base	11	4	7	
others	4	1	3	
Molecular target drug				0.153
with	29	8	21	
without	26	13	13	
Radiation therapy				0.502
RT (+)	11	3	8	
RT (−)	44	18	26	
Histology (biopsy)				0.903
well, moderate	49	19	30	
por, sig, muc	5	1	4	
others	1	1	0	
clinical N status				1
cN (+)	51	20	31	
cN (−)	4	1	3	
clinical M status				0.19
cM (+)	19	10	9	
cM (−)	36	11	25	
clinical Stage				0.462
II	2	0	2	
III	34	11	23	
IV	19	10	9	
Pathological therapeutic effect				0.019
grade 0, 1	36	18	18	
grade 2, 3	19	3	16	

SD, standard deviation.

**Table 2 cancers-11-00635-t002:** FBXW7 staining scores in pre-treatment biopsy specimens and post-treatment resected specimens for each patient.

No	Biopsy Specimen			Resected Specimen			ΔValue *
	Percentage	Intensity	Score	Percentage	Intensity	Score	
1	1	1	1	3	2	6	5
2	1	1	1	3	2	6	5
3	2	1	2	3	2	6	4
4	2	1	2	3	2	6	4
5	2	1	2	3	2	6	4
6	2	1	2	3	2	6	4
7	2	1	2	3	2	6	4
8	2	1	2	3	2	6	4
9	2	1	2	3	2	6	4
10	2	1	2	3	2	6	4
11	2	1	2	3	2	6	4
12	1	2	2	3	2	6	4
13	2	1	2	3	2	6	4
14	3	1	3	3	2	6	3
15	3	1	3	3	2	6	3
16	3	1	3	3	2	6	3
17	3	1	3	3	2	6	3
18	3	1	3	3	2	6	3
19	3	1	3	3	2	6	3
20	3	1	3	3	2	6	3
21	3	1	3	3	2	6	3
22	3	1	3	3	2	6	3
23	2	2	4	3	2	6	2
24	1	1	1	3	1	3	2
25	1	1	1	3	1	3	2
26	1	1	1	3	1	3	2
27	1	1	1	3	1	3	2
28	1	1	1	3	1	3	2
29	1	1	1	3	1	3	2
30	1	1	1	3	1	3	2
31	1	1	1	2	1	2	1
32	1	1	1	2	1	2	1
33	2	1	2	3	1	3	1
34	2	1	2	3	1	3	1
35	1	2	2	3	1	3	1
36	2	1	2	3	1	3	1
37	2	1	2	3	1	3	1
38	2	1	2	3	1	3	1
39	2	1	2	3	1	3	1
40	3	2	6	3	2	6	0
41	3	1	3	3	1	3	0
42	3	1	3	3	1	3	0
43	3	2	6	3	2	6	0
44	3	1	3	3	1	3	0
45	3	2	6	3	2	6	0
46	3	2	6	3	2	6	0
47	3	2	6	3	2	6	0
48	3	1	3	3	1	3	0
49	3	1	3	3	1	3	0

* ΔValue means the value which subtracts FBXW7 staining score of biopsy specimen from that of resected specimen.

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
