# Peer review of "F-Box/WD Repeat Domain-Containing 7 Induces Chemotherapy Resistance in Colorectal Cancer Stem Cells"

_cancers, 2019, doi:10.3390/cancers11050635_

Reviewer 1 Report

The authors seek to understand the molecular mechanism underlying how Fbw7 induces dormancy associated resistance to chemotherapy in colorectal cancer stem cells. The paper is clearly written, however, the following concerns should be addressed before its publication at Cancers.

Figure 1A, it will be nice if the authors could provide experimental evidence validating IHC with Fbw7 antibody, such as probing section of frozen cell pallets derived from WT or Fbw7-/- HCT116 colon cancer cell lines.

Figure 3B, it will be nice if the authors comment whether they only measured the expression of Fbw7alpha isoform or all three isoforms.

Figure 4A, it will be important for the authors to include IB against Fbw7 to validate the knockdown efficiency of shFbw7s.

Author Response

Thank you very much for your kind invitation to submit a revised draft of our manuscript entitled, “F-Box/WD repeat domain-containing 7 induces dormancy-associated chemotherapy-resistance in colorectal cancer stem cells” to Cancers. We also appreciate the time and effort you and each of the reviewers have dedicated to providing insightful feedback on ways to strengthen our paper. Thus, it is with great pleasure that we resubmit our article for further consideration.

We have incorporated changes that reflect the detailed suggestions you have graciously provided. We also hope that our edits and the responses which we provide below satisfactorily address all the issues and concern you and the reviewers have noted.

To facilitate your review of our revisions, attached please find  a point-by-point response to the questions and comments delivered in your letter.

Reviewer 2 Report

The presented study by Shusaku Honma and colleagues deals with the role of FBXW7 in colorectal cancer  therapeutic outcome, CRC stemness and chemoresistance. These factors are of vital interest to a broad range of professionals, as Fbxw7 is the fourth most common mutated tumour suppressor in CRC, and mutations or genetic loss facilitate chemoresistance to standard of care compounds, such as 5-FU.

The study includes patient material, the assessment of FBXW7 protein status by IHC, followed by a retrograde analysis of treatment reponse in relation to FBXW7 status. The observed role is then interrogated in patient PDX samples isolated from mice and analysed for CRC marker expression by FACS and qPCR analysis of stem cell markers. Then, the authors interrogate the ability of FBXW7 to alter the proliferative status of CRC cell lines by using shRNA interference, followed by chemotherapeutic intervention. And finally, a xenograft experiment with the human cell line DLD-1 in mice revealed the increased sensitivity to CPT-11 upon loss of FBXW7.

Separate parts of the manuscript are very interesting, however, it is not presented in a coherent fashion. The parts of the study are not clearly connected and individual observations are not well characterised.

Therefore, concerns arise, which need to be addressed.

Major comments:
Figure 1: An interesting observation is the correlation between poorer treatment response and FBXW7 status. Here the authors report that elevated expression of FBXW7 results in poor prognosis, when compared to low expressing primary tumour samples. This is in contrast to previous studies and available datasets, correlating low FBXW7 expression with poorer overall survival survival. Secondly, FBXW7 is frequently mutated. Did the authors address the mutational status of FBXW7? Also, the classification of FBXW7 expression, ranging from 0 to 12, seems very extreme, given the small amount of samples to start with. Would it be possible to reassess the staining intensity with a score ranging from 0-5? Would effects be lost? Please show representative images in supplementary files for each score.

Figure 2: Please clarify the timeline between concluding the treatment regime and taking samples to reassess the IHC for the highlighted markers. This is very important. Please have similar image panels for all 4 patients shown, including H&E, FBXW7 (low mag and high magnification), Ki67. Please quantify Ki67 on a representative cohort.
Is the recurrence in FBXW7 staining by IHC due to the fact that wild type tissue was biopsied? Are the show images, in particular patient 4, tumour areas? What was the treatment regime of patients shown? Only chemotherapy, as used later for the cell based experiments or xenografts, or chemo-radiotherapy? Please specify.

Figure 3: The presented data is very interesting and one starts to wonder why the xenograft cell lines were not used for the subsequent part of the study, as they can be propagated and analysed. Here, a control tissue sample would be informative. Resected, non-transformed colorectal mucosa tissue could be included and expression of FBXW7, BMII and LGR5 assessed in comparison to the EpCAMhigh/CD44high or EpCAMhigh/CD44low subpopulations. Are H&E from these PDX available? If so, please include with a CD44 IHC staining. Also, would the EpCAMhigh/CD44low cell population be able to re-engraft and from a tumour? Please discuss.

Figure 4 and 5: As FBXW7 is mutant or genetically modified, but rarely transcriptionally down-regulated, it would be more informative to use a FBXW7 CRISPR/Cas9 targeted cell line rather than an shRNA based model. HCT116 cells are readily available with genetic depletion of all FBXW7 isoforms (https://www.ncbi.nlm.nih.gov/pmc/articles/PMC2426948/). Mutant Fbxw7 knock in mice were generated in the past and could be available, too (https://gut.bmj.com/content/63/5/792 and https://www.ncbi.nlm.nih.gov/pubmed/20638938). Furthermore, in these cell lines as well as organoid cultures, other laboratories have interrogated the chemoresistance mediated by loss of FBXW7 (https://www.ncbi.nlm.nih.gov/pmc/articles/PMC4830362/).

Figure 5 C and D: The tumour growth experiment is not clearly defined as a tumour initiation rather then a treatment experiment, as treatment with the CPT-11 compound started a day post engraftment. Usually, one would expect to let tumours engraft until they reach a certain diameter (0,5 to 1cm diameter), followed by either PBS/DMSO or chemotherapy treatment, thereby observing a therapeutic effect. Please reassess the analysis in Figure 5C again and include tumour numbers/ animal.

At the end, it is, from the experimental section, not clear how the quiescent stem cell population described in the beginning is investigated in this study. If the manuscript would aim at the possibility that, instead of genetically depleting or mutating, but rather modifying the expression of FBXW7 provides a therapeutic window, that would be ok. This would also be in line with the majority of already published work in the field, as accelerating tumour cells should make them more vulnerable. However, losing FBXW7 completely, changes cellular responses. This was not discussed.

In the end, an interesting observation, which requires addressing of above mentioned points.

Author Response

Thank you very much for your kind invitation to submit a revised draft of our manuscript entitled, “F-Box/WD repeat domain-containing 7 induces dormancy-associated chemotherapy-resistance in colorectal cancer stem cells” to Cancers. We also appreciate the time and effort you and each of the reviewers have dedicated to providing insightful feedback on ways to strengthen our paper. Thus, it is with great pleasure that we resubmit our article for further consideration.

We have incorporated changes that reflect the detailed suggestions you have graciously provided. We also hope that our edits and the responses which we provide below satisfactorily address all the issues and concern you and the reviewers have noted.

To facilitate your review of our revisions, attached please find a point-by-point response to the questions and comments delivered in your letter.

Round  2

Reviewer 2 Report

No further comments.